# Research on Real-Time Monitoring and Warning Technology for Multi-Parameter Underground Debris Flow

Qingtian Zeng [1], Sitao Zhu [2,*], Zhengrong Li [1], Aixiang Wu [2], Meng Wang [2], Yan Su [1], Shaoyong Wang [3], Xiaocheng Qu [3] and Ming Feng [3]

[1] Yunnan Diqing Nonferrous Metals Co., Ltd., Diqing 674400, China
[2] School of Civil and Resource Engineering, University of Science and Technology Beijing, Beijing 100083, China
[3] Beijing Anke Xingye Technology Co., Ltd., Beijing 100083, China
* Correspondence: zhusitao@ustb.edu.cn

**Abstract:** Sudden debris flows in underground mines are characterized by strong burstiness, great destructiveness, and difficult monitoring. Traditional single monitoring methods can only roughly judge the probability of underground debris flow occurrences through one-sided potential phenomena, making it difficult to accurately predict sudden underground debris flows. Therefore, effective monitoring methods can prevent or reduce waste and damage to mineral resources caused by mine debris flow disasters. This study is based on the theoretical foundations of rainfall automatic identification program, unsteady flow theory, and wavelet threshold denoising theory. It preprocesses key data such as rainfall, groundwater, and surface displacement with the aim of reducing criterion errors and improving the accuracy of determination. By utilizing the underground debris flow warning determination program, warning determination algorithm, and information management system hosted on the monitoring and warning platform, a comprehensive underground debris flow warning system is integrated. This system incorporates determining parameters such as rainfall, water inflow, groundwater level, surface subsidence, pore water pressure, surrounding rock stress, microseismic phenomena, and underground video recognition, with the innovative approach of "weather-surface-underground" multi-directional monitoring. The system was successfully installed and applied in the Pulang Copper Mine in Yunnan Province, demonstrating good application effectiveness. The results indicate that compared to traditional single monitoring methods, the multi-directional monitoring and warning system for underground debris flows has advantages such as low fault tolerance and high accuracy, making it more suitable for ensuring safe mining in mining areas.

**Keywords:** data platform; debris flow; early warning system; multi-parameter

## 1. Introduction

The natural caving mining method, a cost-effective and large-scale mining approach, has been extensively utilized in underground mining within the mining industry. This method relies on the natural collapse of the rock mass under the influence of self-weight stress and tectonic stress. The collapsed rock is then transported through the ore pass at the mining level, resulting in gradual surface subsidence and the formation of open pits. During the summer season, various materials such as fragmented rock, soil in the subsidence pit, glacial deposits, loose accumulations around the mining area, and fractured geological structures serve as the source materials for debris flow formation. The occurrence of rainfall and snowmelt acts as the water source and triggers the driving force for debris flow formation. Moreover, the presence of geological structures like joints and faults in the overlying rock layers of the ore body leads to continuous loosening and fragmentation of these layers, ultimately causing debris flow disasters. Underground debris flows are characterized by their sudden occurrence, intense momentum, and high destructive power. To effectively prevent and control the occurrence of underground debris flows, it is crucial

to implement measures that consider the interrelationships and characteristics of the factors involved in debris flow formation. Currently, the primary methods for preventing and controlling underground debris flows in the mining industry include the following: (1) installing drainage facilities in the surrounding areas to manage water flow; (2) strengthening weather monitoring and forecasting to anticipate potential debris flow events; (3) timely sealing of abandoned tunnels and chutes to prevent the entry of loose materials; (4) controlling the mining output to ensure uniform extraction in each mining panel, thereby minimizing the potential for destabilizing the surrounding rock mass; (5) developing emergency plans specifically tailored for dealing with underground debris flow incidents.

The causes of underground debris flows can be categorized into two main groups: gradual factors and sudden factors. Gradual factors encompass geological conditions, topographical conditions, and tectonic activities, while sudden factors include elements like blasting vibrations and rainfall [1]. Traditional monitoring methods typically prioritize rainfall monitoring since it is one of the primary influencing factors for underground debris flows. However, relying solely on monitoring rainfall intensity is inadequate for accurately determining the occurrence of underground debris flows. To enhance the accuracy and timeliness of early warning systems, it is crucial to implement comprehensive monitoring and integrated determination of both gradual and sudden factors.

In comparison, the causes of fault or goaf debris flows in coal mines involve the intrusion of material from fault zones or fragmented rock debris into the working space under the influence of groundwater, while debris flows in metal mines occur when surface soil layers or fragmented rocks enter the working space through fractures and galleries formed by surface water-induced caving mining. Therefore, debris flows in metal mines are more easily predicted in advance through monitoring and early warning methods. Rainfall is the decisive factor in triggering debris flows. Tan et al. [2] selected the rainfall intensity during the 10 min period of debris flow generation and the daily precipitation for combined analysis and proposed that the number of debris flows can be calculated by using the boundary values of daily maximum rainfall and heavy rainfall. Pang et al. [3] evaluated debris flow disasters under different rainfall frequencies through numerical simulation, revealing the characteristics of the occurrence process of debris flow disasters induced by short-term heavy rainfall. Liang et al. [4], Ni et al. [5], and Zhao et al. [6] conducted research on the contribution of preliminary rainfall to rainstorm-induced debris flows and the prediction model of debris flow early warning, and proposed an early warning framework and recommendations for rainfall-induced debris flows. Pan Huali et al. [7] proposed a method to calculate the threshold rainfall for debris flow early warning based on the hydraulic initiation mechanism of debris flows, as the current empirical methods and frequency calculations are unable to meet the needs of debris flow early warning.

The research on the formation mechanism of debris flows is still in its early stages. Iverson predicted the occurrence of debris flows based on the changing trends of internal physical properties of the soil, such as water content and pore water pressure [8]. Eckersley et al. suggested that pore water pressure rapidly increases when soil undergoes failure [9]. After conducting an analysis of the ultimate equilibrium of sediment-laden flows in saturated channel beds, Takahashi determined the critical initiation conditions for debris flows in saturated channel bed deposits under surface water flow [10]. Hungr O et al. analyzed the process and conditions of debris flow formation and concluded that the occurrence of debris flows is caused by the rapid reduction in matric suction of source material and accelerated shear deformation [11].

Currently, the monitoring and early warning techniques for debris flows are not yet mature, and the main methods include contact-type warning devices and non-contact-type warning devices [12–14]. With the widespread application of new-generation information technology, many scholars have successfully applied debris flow early warning systems developed based on B/S or C/S architecture [15–18]. Currently, both domestically and internationally, the research focus on debris flow early warning is mostly on open-pit mines,

while the research on underground debris flows is relatively scarce and limited in scope, lacking comprehensive consideration of multi-directional and multi-factor influences.

This paper aims to develop an underground debris flow monitoring and early warning system that incorporates multi-factor monitoring. It focuses on the preprocessing of monitoring parameters and explores the creation of a real-time warning model based on the "weather-surface-underground" tripartite monitoring approach. The integration with the monitoring platform is pursued to enable real-time and precise early warning capabilities.

## 2. Critical Data Preprocessing

The terms "Weather," "Surface," and "Underground" refer to atmospheric monitoring, geodetic monitoring, and downhole monitoring, respectively. Multi-parameter monitoring is conducted in each of these domains, and the collected data are preprocessed before being transmitted to the big data platform. The system performs early warning processing by comprehensively evaluating the received data, as illustrated in Figure 1.

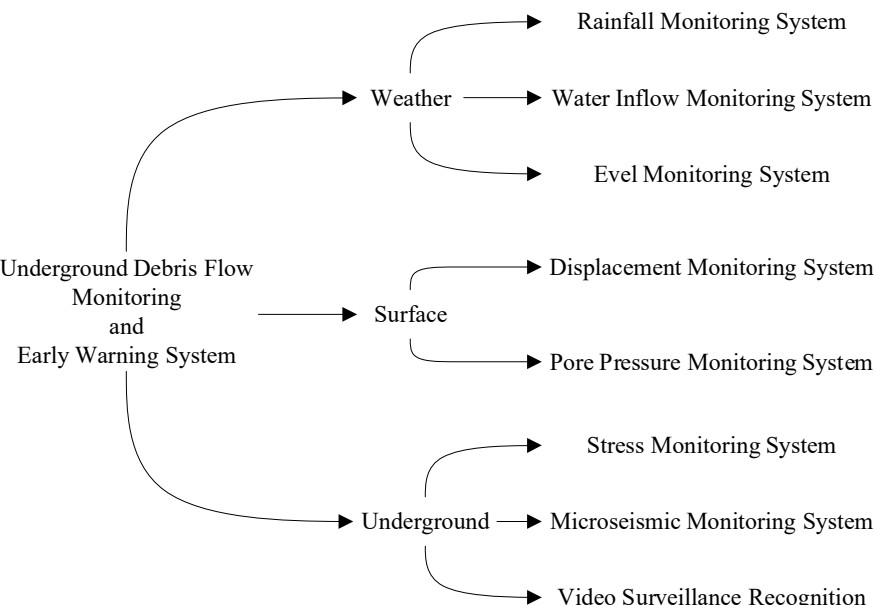

**Figure 1.** Monitoring and early warning system structure.

In the context of underground debris flow, rainfall and groundwater are the primary influencing factors, while surface displacement is the main observable phenomenon. Monitoring data serve as the foundation of the entire early warning system. The use of reasonable and effective monitoring data can significantly enhance the accuracy of monitoring and early warning. Additionally, scientific preprocessing of raw data plays a crucial role in highlighting the characteristics of the data.

### 2.1. Research on Rainfall Monitoring Identification

The key to the accuracy of rainfall monitoring is how to judge the effective rainfall time and rainfall intensity. The key to determining the effective rainfall time is establishing how to identify the starting point and ending point of rainfall. Scholars at home and abroad have conducted much research on the division of rainfall events [19–24]. This paper chooses the following effective rainfall time identification mark proposed by Jan et al. [23]:

1. The time when the hourly rainfall is greater than 4 mm in a continuous rainfall is taken as the starting time;
2. Six hours of continuous rainfall of less than 4 mm is taken as the end time.

In reality, the real-time warning process is often in the process of rainfall, so the starting time of the rainfall process should be obtained every time the rainfall intensity

is calculated. If there is no rainfall, the current time is returned; otherwise, the rainfall monitoring data are traversed forward to obtain the starting time of rainfall. He et al. [22] stated that according to the effective rainfall starting time mark, the time point that meets the conditions of 0 rainfall in the first 3 h and less than 4 mm rainfall in the first 6 h is selected as the starting time point of rainfall, and the starting time of the rainfall is quickly found in each calculation process (Figure 2).

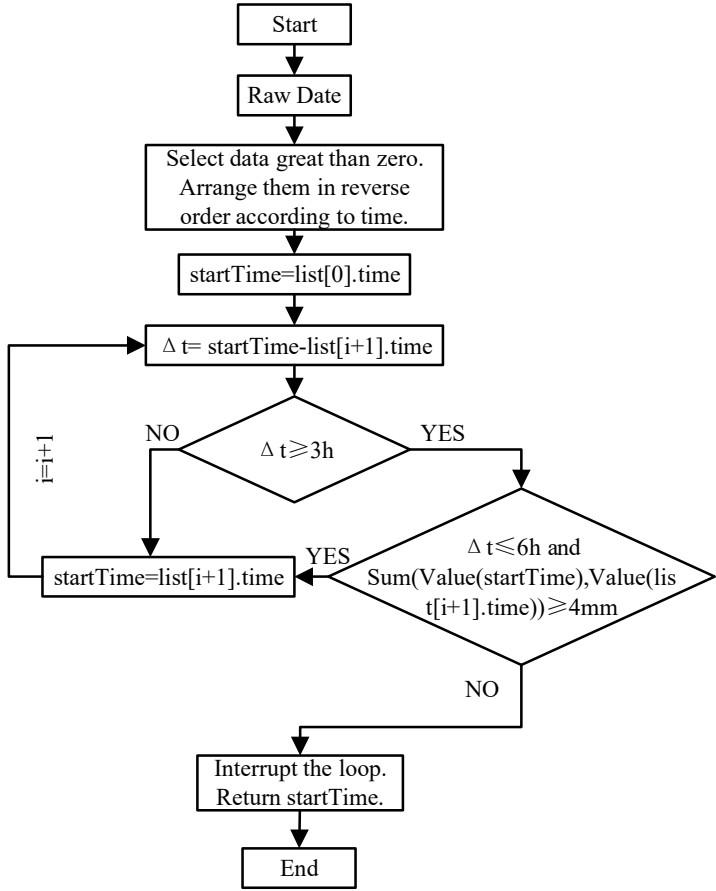

**Figure 2.** The process of obtaining the starting point of effective rainfall time.

By using the identification mark of the start time of rainfall, the rainfall intensity and cumulative rainfall can be calculated within 10 min intervals. This calculation is performed using the data collected by the rainfall recorder during the period of effective rainfall.

### 2.2. Research on Groundwater Level Prediction

In mining areas, the mine strata and terrain can be highly complex, leading to anisotropic characteristics of aquifers in different regions. Setting up hydrological monitoring points throughout the entire mining area can be resource-intensive and costly. However, it is possible to predict the hydrological information of the area by strategically placing hydrological monitoring points in specific areas. By carefully selecting representative monitoring points and analyzing the data collected from those points, it is possible to gain insights into the hydrological conditions of the broader mining area. This approach can help optimize resource allocation and provide valuable information for hydrological predictions in the mining environment.

The theoretical basis of deep well precipitation is derived from the theory of groundwater dynamic seepage. The development of groundwater seepage theory can be divided into two stages: the first stage is the generation and development of groundwater steady flow theory, and the second stage is the generation of groundwater unsteady flow theory.

In the middle of the 19th century, Henry Darcy and J. Dupuit conducted studies on one-dimensional steady flow and two-dimensional steady movement of flowing wells. Their work laid the foundation for the theory of steady seepage of groundwater. However, the steady flow theory alone cannot fully describe the entire development process of groundwater movement, and the practical application of steady seepage theory has significant limitations. In the early 20th century, C. V. Theis studied the unsteady flow of confined groundwater wells. He specifically investigated the unsteady seepage flow of groundwater caused by pumping from a single well in a homogeneous, isotropic, equally thick, laterally infinite, non-rechargeable horizontal aquifer. The result was the formulation of the famous Theis equation, which marked the beginning of a new era in modern groundwater movement theory.

The Theis equation provides a mathematical framework to analyze the unsteady flow of groundwater in response to pumping from a well. It allows for the estimation of parameters such as aquifer transmissivity and storativity. This development has greatly enhanced our understanding of groundwater dynamics and has been widely applied in hydrogeology and water resources management.

This study uses the analysis method of unsteady flow to predict the inflow and groundwater level [25]. Under the assumption of no interference from neighboring wells, the mathematical model for single-well pumping in an aquifer makes the following assumptions:

1.  The aquifer is homogeneous, isotropic, of uniform thickness, laterally infinite, and horizontally layered.
2.  The natural hydraulic gradient is zero before pumping.
3.  Pumping is conducted at a constant discharge rate, and the well diameter is infinitesimally small.
4.  Flow in the aquifer follows Darcy's law.
5.  The release of groundwater storage caused by the decline in hydraulic head occurs instantaneously.

After pumping, a cone of depression is formed with the well axis as the axis of symmetry, and the well axis serves as the z-axis (Figure 3), with the origin of the coordinate system located at the bottom of the aquifer along the z-axis. At this point, the fully penetrating well flow mathematical model for the top discharge rate of a single well is given by:

$$\begin{cases} \dfrac{\partial^2 s}{\partial r^2} + \dfrac{1}{r}\dfrac{\partial s}{\partial r} = \dfrac{\mu}{T}\dfrac{\partial s}{\partial t} & (t > 0, 0 < r < \infty) \\[2mm] s(r,0) = 0 & (0 < r < \infty) \\[2mm] s(\infty,t) = 0, \ \dfrac{\partial s}{\partial r}\Big|_{r\to\infty} = 0 & (t > 0) \\[2mm] \lim\limits_{r\to 0} r\dfrac{\partial s}{\partial r} = -\dfrac{Q}{2\pi T} \end{cases} \tag{1}$$

In the equation, $s = s(r, t) = H_0 - H$.

The solution to Equation (1) is:

$$s = \frac{Q}{4\pi T}W(u) \tag{2}$$

$$\begin{cases} W(u) = \int_u^\infty \dfrac{e^{-y}}{y}dy \\[2mm] u = \dfrac{r^2\mu}{4Tt} \end{cases} \tag{3}$$

where $W(u)$ is the Theis well function; $s$ is the water level drop at any time at any point within the influence range of pumping; $Q$ is the flow rate of pumping wells; $T$ is the coefficient of transmissivity; $t$ is the time from the beginning of pumping to the calculation time; $r$ is the distance from the calculation point to the pumping well; $\mu$ is the water storage coefficient of the aquifer.

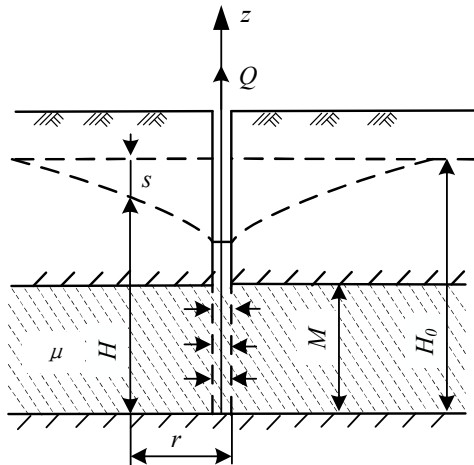

**Figure 3.** Schematic diagram of pumping at measuring points.

Expand $W(u)$ in Equation (3) as a series to obtain Equation (4):

$$W(u) = \int_u^\infty \frac{1}{y} e^{-y} dy = -0.577216 - \ln u + u - \sum_u^\infty (-1)^n \frac{u^n}{n \cdot n!} \tag{4}$$

It can be seen from this that the series after the first three terms of the $W(u)$ expansion is an alternating series. The sum of the alternating series does not exceed $u$, so the error of replacing $W(u)$ with the first two terms of the expansion will not exceed $2u$. Generally, the relative error in production is allowed to be about 5%. Then, when $u \le 0.05$ (instant), the Theis well function can be replaced by the first two items. Thus, $W(u)$ can be represented by Equation (5):

$$W(u) = -0.577216 - \ln u = \ln \frac{2.25Tt}{r^2 \mu} \tag{5}$$

Therefore, Equation (2) can be approximated by the Jacob formula, as shown in Equation (6):

$$s = \frac{Q}{4\pi T} \ln \frac{2.25Tt}{r^2 \mu} = \frac{0.183Q}{T} \lg \frac{2.25Tt}{r^2 \mu} \tag{6}$$

By rearranging Equation (6), we obtain the expression of flow rate as a function of drawdown:

$$Q = \frac{sT}{0.183 \cdot \lg[(2.25Tt)/(r^2\mu)]} \tag{7}$$

The above is deduced under the assumption that the flow is fixed and the flow is usually seasonal, as shown in Figure 4:

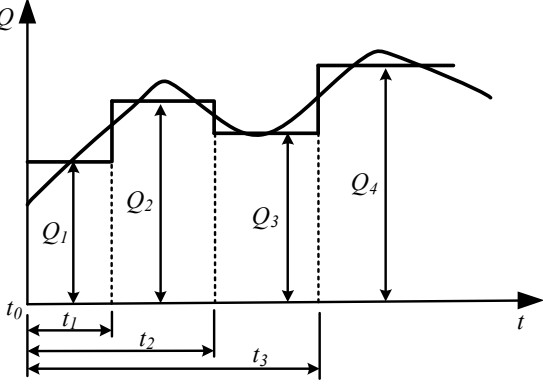

**Figure 4.** Flow rate conceptualization change chart.

Equation (7) is derived under the assumption of a constant and unchanging flow rate. However, in reality, flow rates typically exhibit seasonal variations, as illustrated in Figure 4.

In this situation, the flow rate profile can be approximated by a stepwise piecewise-linear function. Each step difference can be considered as a constant flow rate. By applying Equation (7) and using the superposition principle, the drawdown caused by each step difference in flow rate can be summed up, resulting in the drawdown equation for varying flow rates.

$$s = \frac{1}{4\pi T} \sum_{i=1}^{n} (Q_i - Q_{i-1}) W \left[ \frac{r^2 \mu}{4T((t-t_{i-1}))} \right] \quad t_{i-1} < t < t_i \tag{8}$$

Equation (8) can be approximated using the Jacob's formula as follows:

$$s = \frac{0.183}{T} \sum_{i=1}^{n} (Q_i - Q_{i-1}) \lg \frac{2.25T(t - t_{i-1})}{r^2 \mu} \tag{9}$$

where $Q_i$ and $Q_{i-1}$ are the generalized flow values of $t_i$ and $t_{i-1}$ in time.

*2.3. Research on Preprocessing of Surface Displacement Data*

The long-term monitoring curve reflecting surface displacement often exhibits jagged mutations, where the rate of change suddenly increases at a certain time but stabilizes afterward. This fluctuation phenomenon in the monitoring curve can be attributed to both natural activities such as rainfall and earthquakes and human activities like blasting and construction. Additionally, factors such as unstable voltage of electrical signals, electromagnetic field interference, and insufficient instrument shielding can introduce noise into the monitoring signal, further contributing to the fluctuation phenomenon.

To improve the accuracy of evaluating real surface displacement, it is necessary to denoise the obtained monitoring signal. This denoising process aims to eliminate the noise component while effectively retaining the sudden changes caused by accelerated surface deformation. By removing the noise, a clearer and more reliable signal can be obtained, enabling a more accurate assessment of slope stability in real time.

Achieving real-time and accurate evaluation of slope stability requires a careful balance between noise elimination and preserving relevant deformation information. Advanced signal processing techniques, such as filtering algorithms and statistical methods, can be employed to achieve this goal. These techniques help to reduce the impact of noise while maintaining the important features of the monitoring signal associated with surface deformation.

The surface subsidence data obtained by long-term monitoring cause large fluctuations in the monitoring process due to human operation, instrument accuracy, or environmental changes. Therefore, this part of the data contains two parts: true value and noise. It is necessary to perform noise reduction preprocessing before monitoring data transmission. Wavelet threshold denoising is mostly used for data denoising, and its effect depends on the influence of the threshold function [24]. The basic steps of wavelet threshold denoising are as follows:

1.  The original signal is decomposed, and the low-resolution scale coefficients and wavelet coefficients at each resolution are obtained by using the orthogonal wavelet transform algorithm.
2.  According to the threshold selection rules, the invalid noise is removed, the effective signal is extracted, and the high-frequency effective signal is retained.
3.  Reconstruction of wavelet coefficients. All low-frequency scale coefficients and the obtained wavelet coefficients are reconstructed to obtain the estimation of the original data.

The hard threshold function and soft threshold function are two commonly used threshold methods for wavelet denoising:

(1) Hard threshold function

$$\widehat{\omega}_{j,k} = \begin{cases} \omega_{j,k} & \left|\omega_{j,k}\right| \geq \lambda \\ 0 & \left|\omega_{j,k}\right| < \lambda \end{cases} \tag{10}$$

(2) Soft threshold function

$$\widehat{\omega}_{j,k} = \begin{cases} sign\left(\left|\omega_{j,k}\right| - \lambda\right) & \left|\omega_{j,k}\right| \geq \lambda \\ 0 & \left|\omega_{j,k}\right| < \lambda \end{cases} \tag{11}$$

In the formula, $\widehat{\omega}_{j,k}$ is the threshold function, $\lambda$ is the selected threshold, $\omega_{j,k}$ is the transformed wavelet coefficients.

The difference between the two methods lies in how they handle the transformed wavelet coefficients during the denoising process. The hard threshold function forces the wavelet coefficients to either zero or non-zero values, effectively removing coefficients below a certain threshold. On the other hand, the soft threshold function optimizes the convergence of the retained wavelet coefficients by shrinking the magnitude of coefficients towards zero.

Numerous practical applications have shown that data processed using the soft threshold function tend to result in smoother and more regular outcomes. In this research paper, the focus is on noise reduction, and therefore, the soft threshold processing method is chosen. By applying the soft threshold function, the denoising process aims to retain important information while reducing the impact of noise, resulting in a smoother and more reliable dataset for further analysis.

## 3. Early Warning Judgment of Underground Debris Flow

With the rapid development of communication networks and Internet of Things technology, non-engineering control measures represented by debris flow monitoring and early warning systems have begun to appear. The debris flow early warning and monitoring system is generally composed of a field monitoring network and a monitoring and early warning software platform. The field monitoring network mainly monitors the physical characteristics of debris flow in real time by using various advanced instruments and equipment and transmits the information regularly or irregularly to the monitoring and early warning software platform of the monitoring center through the remote infinite network. The system sends an early warning signal after data analysis.

### 3.1. Early Warning Procedures

The underground debris flow monitoring and early warning system can greatly improve the information level of early warning work, improve the accuracy of early warning, facilitate the formulation of efficient emergency measures, minimize the loss of life and property caused by underground debris flow disasters, and ensure the safety of people's lives and property. The debris flow warning program (Figure 5) mainly includes the following aspects:

1.  Determination of the main factors and mechanisms of underground debris flow formation.

The factors affecting underground debris flow are determined based on the surface conditions of the mining area, the distribution of geological strata, the distribution of glacial deposits in the mining area, and the mining methods. Since rainfall within a mining area often exhibits consistency, rainfall is not considered a differentiating factor for different locations within the same mining area.

2.  Monitoring of factors inducing underground debris flow.

Targeted monitoring is implemented based on the identified factors influencing underground debris flow. Rainfall intensity monitoring points can be established in areas of the mining area unaffected by mining activities. The amount of water inflow in the mine is influenced by various factors such as hydrological characteristics, climatic influences, geomorphological features, structural faults, and lithology of geological strata. Therefore, water inflow monitoring points need to be strategically placed according to different situations, such as in areas with structural faults or poor lithology. Surface displacement monitoring, groundwater monitoring, and pore water pressure monitoring primarily focus on subsidence areas and areas with significant deformation, with particular emphasis on monitoring the central locations of subsidence areas and their surroundings. Stress monitoring, microseismic monitoring, and video surveillance are mainly aimed at monitoring stress changes in surrounding rock masses and the condition of minerals during underground mining. Video surveillance can identify mineral moisture content and fragmentation conditions based on mineral brightness, chromaticity, and edge lines of rock masses.

3. Warning procedures.

After preprocessing the monitoring data of factors inducing underground debris flow using specific algorithms, the backend decision-making system determines whether to trigger a warning based on the processed data and the warning determination algorithm. If the warning threshold is reached, the warning information management system determines the affected objects and warning levels based on the category and values of the monitoring data that triggered the warning and then publishes the warning information.

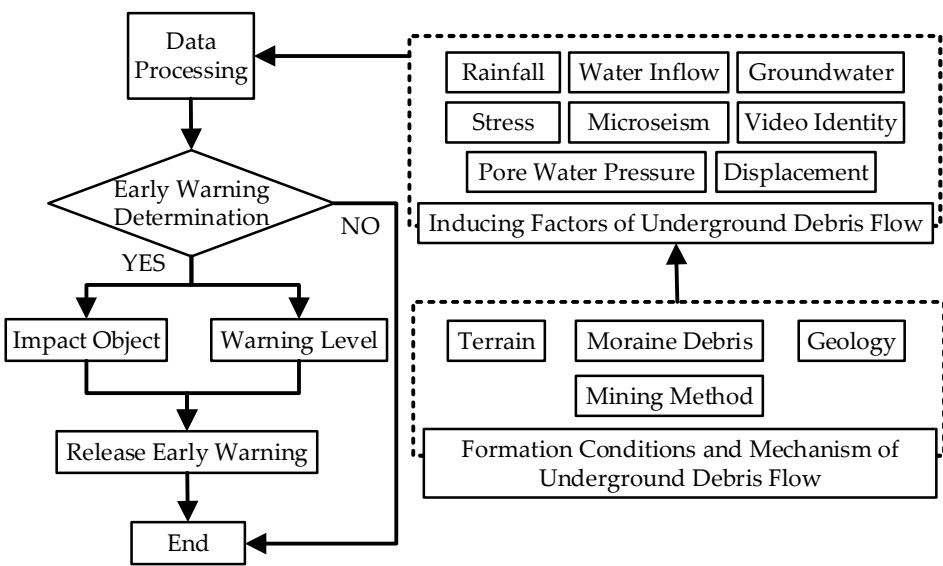

**Figure 5.** Underground debris flow monitoring and warning flowchart.

*3.2. Early Warning Decision Algorithm*

Taking disaster warning as the target layer of the structural model, in the establishment of the evaluation index system, the indicators are divided into three categories: rainfall, surface, and underground are used as the criterion layer; rainfall, water inflow, water level, settling volume, pore water pressure, surrounding rock stress, microseism, and video recognition are used as index layers.

The early warning index is determined by the index method, and the early warning index ($I_{SI}$) is determined by the combination of a single index and a comprehensive index ($I_{CI}$). The early warning determination algorithm is shown in Figure 6. $I_R$, $I_E$, and $I_U$ are weather monitoring indicators, surface monitoring indicators, and underground monitoring indicators, respectively, which are calculated by their corresponding monitoring target data and weights (such as $K_{r1}\sim K_{r3}$, $K_{w1}\sim K_{w2}$, $K_{j1}\sim K_{j3}$). In the single index method, $I_{SI}$ is the maximum value in $I_R$, $I_E$, and $I_U$. In the comprehensive index, it is the weighted

average of the weather monitoring index, surface monitoring index, and underground monitoring index, and $K_1 \sim K_3$ is the weight. The weight can be determined according to the analysis of the influencing factors of debris flow disasters in the region and the activity of monitoring indicators so as to improve the adaptability of the early warning algorithm to the complex environment.

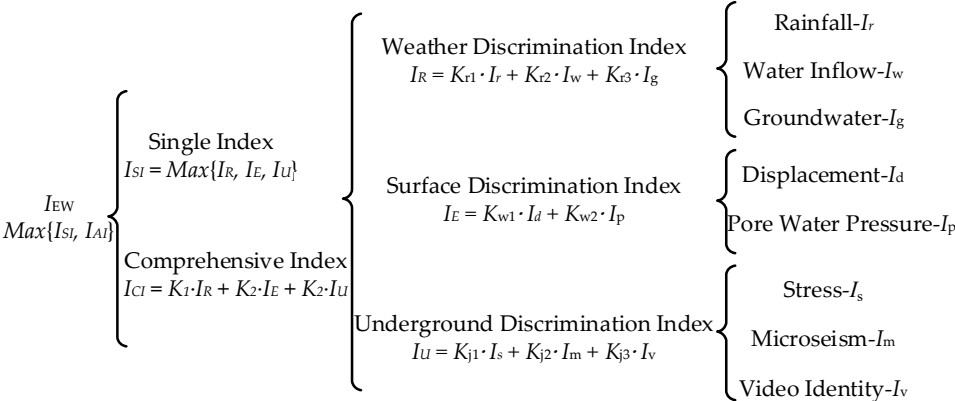

**Figure 6.** Structure diagram of early warning judgment algorithm.

Based on the magnitude of the warning index, it is divided into four risk levels: "no danger, low risk, moderate risk, and high risk". Risk assessment is conducted based on the actual conditions at the site, as shown in Table 1.

**Table 1.** Level 4 warning and corresponding prevention and control suggestions for the monitoring area platform.

| Warning Level | Degree of Danger | Risk Assessment |
|---|---|---|
| I | No danger | Normal mining |
| II | Weak danger | Feasible for mining |
| III | Medium danger | Reduced mining intensity |
| IV | Strong danger | Sequentially releasable for mining |

When the warning is determined to have no danger (Level I), normal operations can continue in the work area. When the warning is determined to be of low risk (Level II), monitoring and warning attention should be strengthened, and professionals should be dispatched to assess the warning. When the warning is determined to be of moderate risk (Level III), it is necessary to strictly control the number of personnel in the warning area and adjust the mining intensity accordingly. Professionals should also be sent to the warning area to verify the situation. When the warning is determined to be of high risk (Level IV), immediate cessation of operations is required. Site personnel should be evacuated, and the next step of the risk mitigation plan should be implemented.

### 3.3. Early Warning Information Management

The warning information management system is one of the core functions of the monitoring and warning system. Its main functions include identifying and calculating warning indices, setting the weights of warning indices, determining the warning levels, storing mining monitoring models, and issuing warning information. Figure 7 illustrates the operational flowchart of the warning information management system.

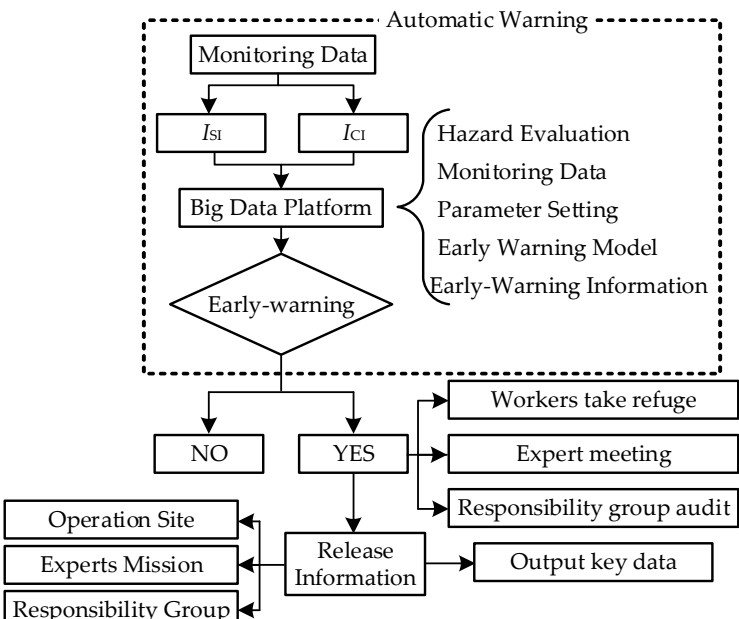

**Figure 7.** Early warning information management system.

Upon receiving monitoring data from various points in the mining area, the on-site warning information management system first identifies the monitoring point to which the data belong. Then, it uses specific algorithms to calculate the warning index and makes a decision on whether to issue a warning based on the judgment. If the system determines that a warning should be issued, the warning information is first conveyed to on-site personnel facing immediate danger through short messages or AI phone calls. Upon receiving the information, on-site personnel take emergency measures to ensure their safety. Simultaneously, the same information is sent to the mine safety responsibility group, which dispatches professionals to verify the warning information and monitoring data, as follows:

1.  If the monitoring data show abnormalities and it is confirmed that there is no danger to on-site personnel, the professionals will go to the monitoring point that triggered the warning to verify the situation.
2.  If the monitoring data show no abnormalities, once the responsibility group confirms the occurrence of a warning, the warning information is sent to members of the expert group. Additionally, the data that triggered the warning are automatically generated and sent to the expert group in the form of images and text. The responsibility group is responsible for organizing expert consultations.

During the mining process, if only a single monitoring alarm is triggered, the alarm information management system will only send the alarm information to the mine responsibility group, and the data will be verified by dedicated personnel.

## 4. Application of Underground Debris Flow Monitoring and Early Warning System in Pulang Copper Mine

### 4.1. Project Background

The Pulang copper mine is located in Shangri-La City, Yunnan Province, China. It is currently the largest underground metal mine in China and operates as a natural caving copper mine on a large scale. Since its construction and production, the mine has achieved good economic benefits but has also experienced multiple underground debris flow accidents. The frequent occurrence of underground debris flows has led to an increase in the dilution rate of extracted ore, an increase in safety hazards, and a decrease in the economic efficiency of the mine. The overall topography of the mining area slopes westward, with higher elevation in the east and lower elevation in the west, ranging from 3600 to 4500 m

above sea level. The slope gradient in the mining area ranges from 25 to 35 degrees, with some local sections being steep and forming cliffs. The mining area has a cold temperate climate, with an average annual temperature of 4 °C. The average temperature in the hottest month is around 10 °C, while in the coldest month, it is around −8 °C. The climate is relatively mild from May to October, and snow accumulation occurs from November to April of the following year. The average annual rainfall in the area is 619.9 mm, with the rainy season (May to October) accounting for 87.1% of the annual rainfall.

As shown in Figure 8, the Pulang copper mine adopts an underground natural caving mining method, where the underground ore body is extracted, resulting in the formation of goaf. This leads to surface movement, subsidence, and collapse, forming large-scale surface collapse pits (Figure 9). The overlying Quaternary glacial deposits and surface water gradually converge with groundwater towards the bottom of the collapse pits. During the transportation process, the glacial deposits are progressively sorted under the action of surface water, with coarse particles being retained along the way, while fine particles reach the bottom of the collapse pits first and form a relatively impermeable layer, causing a significant accumulation of surface water and groundwater at the bottom of the collapse pits.

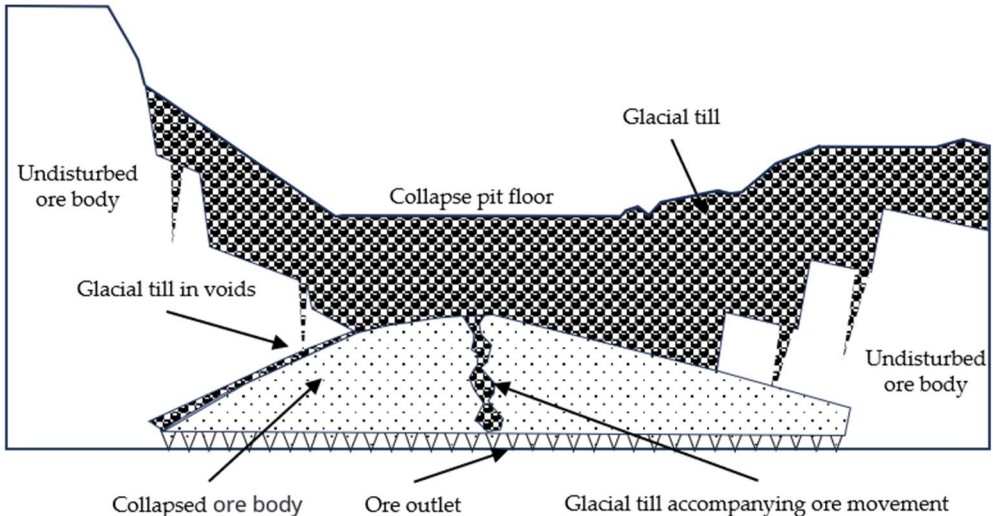

**Figure 8.** Underground debris flow hazard schematic diagram.

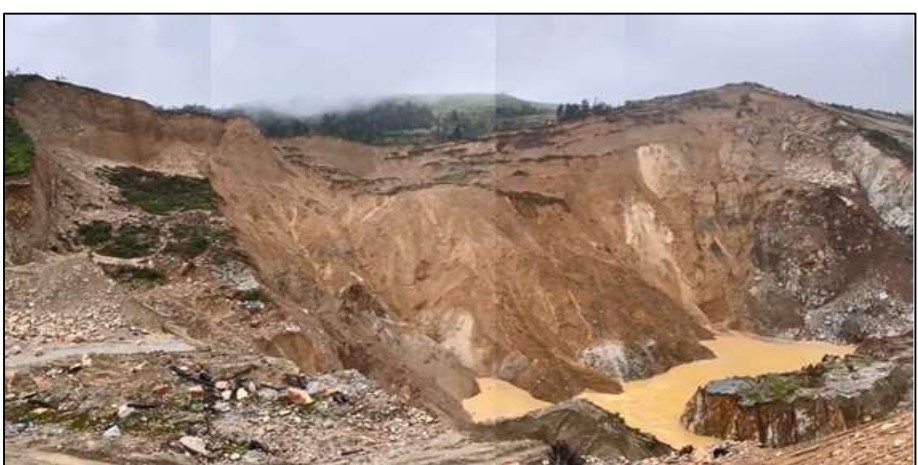

**Figure 9.** On-site photos of surface subsidence.

Part of the mud–water mixture gathered at the bottom of the pits enters the fragmented ore blocks under the influence of gravity and gradually moves downward with the ore body, eventually surging out through the mining opening to form underground debris flows. Some of the mud–water mixture flows out through fault fractures connected to the mining opening or gaps between collapsed and uncollapsed ore bodies, forming underground debris flows.

This study focuses on the implementation of a monitoring and early warning system for the 3720 level of the Pulang copper mine. Building upon the previously described underground debris flow monitoring and early warning system, the mining area is subjected to comprehensive "weather-surface-underground" monitoring and early warning through the deployment of on-site sensors and the installation of an early warning information management system.

### 4.2. Specific Implementation Plan

### 4.2.1. Build the Monitoring Platform

The establishment of the early warning platform consists of three parts: (1) establishing a digital model for monitoring and early warning in the mining area; (2) zoning the underground monitoring and early warning areas; (3) deploying on-site monitoring and early warning sensors.

The digital model of the mining area, as shown in Figure 10, serves as the fundamental component of the monitoring and early warning system. This model is created through on-site exploration and digital modeling techniques and is subsequently imported into a big data platform for analysis and visualization. The digital model of the mining area comprises three main components: the sky model, the ground model, and the underground model. The sky model provides real-time synchronization to display the impact of weather conditions such as rainfall, clear skies, or cloudy weather. This feature enhances the monitoring and early warning platform's display interface, making it more intuitive for users. The ground model combines images with data using unmanned aerial photography technology. This approach allows for a more visually appealing and intuitive representation of the mining area. By integrating aerial imagery with relevant data, the ground model provides a comprehensive view of the mining site, aiding in monitoring and analysis. The underground model is established based on the specific mining conditions and mining plans of the area. This model takes into account the geological characteristics, mining methods, and infrastructure of the underground mine. By incorporating this information, the underground model provides insights into the subsurface conditions, allowing for a more comprehensive understanding of the mining operations.

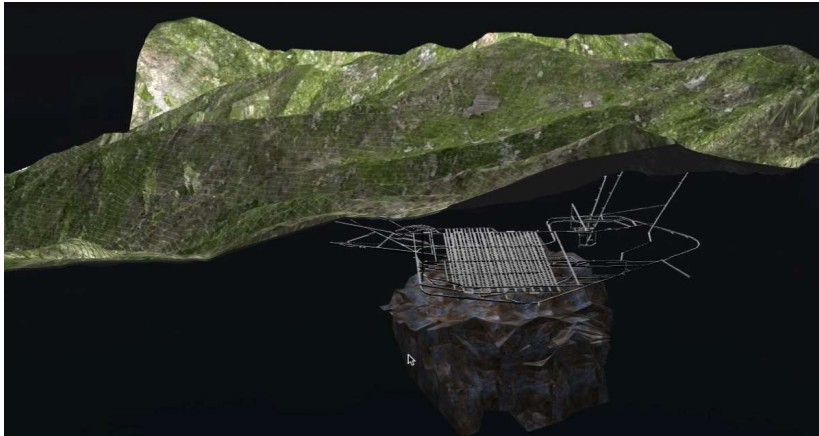

**Figure 10.** Mining area detection model.

Overall, the digital model of the mining area plays a crucial role in the monitoring and early warning system. It facilitates real-time visualization, integrates aerial imagery with data, and incorporates underground mining information, providing a comprehensive and intuitive representation of the mining site for effective monitoring and decision-making.

As shown in Figure 11, the 3720 m mining face has been divided into six monitoring and early warning zones (Area1~Area6) based on the size of the mining area. The entire mining area consists of 13 tunnels (N1~N4 and S1~S9, arranged from north to south).

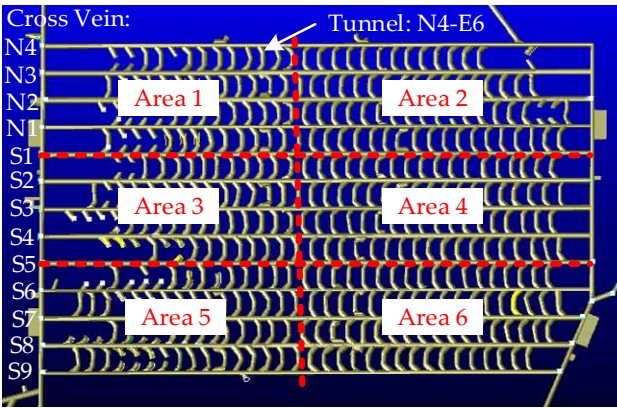

**Figure 11.** Schematic diagram of the 3720 working face.

Figure 12 is a schematic diagram of the display interface of the monitoring and early warning platform, showing some of the on-site sensor installation points. The monitoring points include rainfall monitoring points, hydrological monitoring points, and displacement monitoring points. The function of the rainfall monitoring points is to monitor the rainfall in the mining area and upload the rainfall data to the early warning information management system. The function of the hydrological monitoring points is to monitor changes in groundwater levels and pore water pressures and upload the data to the early warning information management system. The main function of the displacement monitoring points is to monitor surface subsidence in the subsidence area of the mining area and upload the data to the early warning information management system.

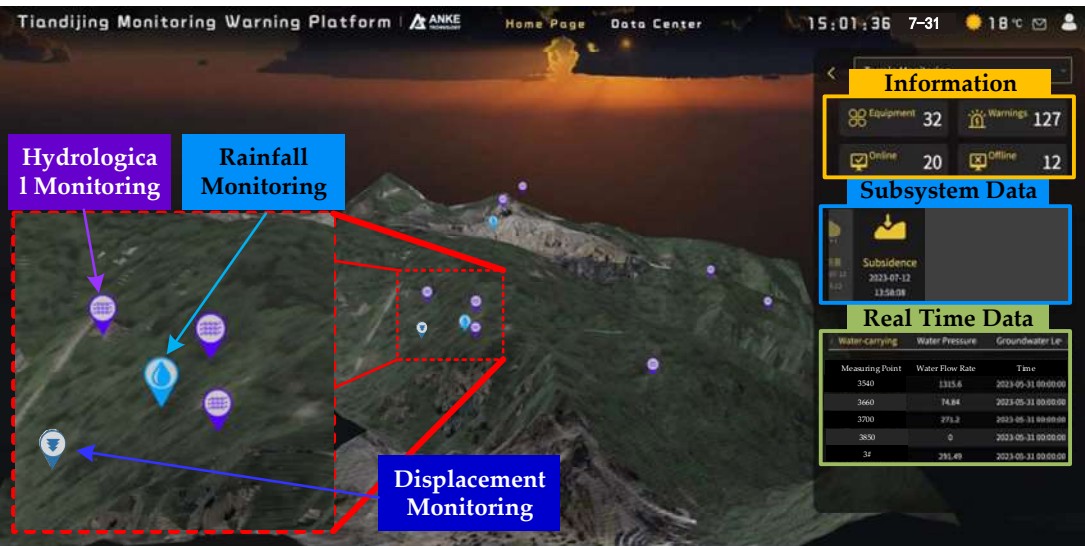

**Figure 12.** Surface and hydrological monitoring points.

In Figure 12, the right side of the warning platform display interface features an information display window that is organized in a top-to-bottom order. This window consists of three sections: the overall information window, the subsystem data, and the real-time data. The overall information window provides an overview of the system's status. It displays the total number of sensors in the system, as well as the number of sensors that are currently online and offline. Additionally, it shows the cumulative number of warning events that have occurred since the system was put into operation. This section provides a high-level summary of the system's performance and alerts the user to any potential issues. The subsystem data section is designed to display and query the monitoring data of each sensor. It provides a list of monitoring items or categories, which could be specific areas or aspects of the mining site. By clicking on a particular monitoring item in this section, the user can access detailed information related to that item.

Upon selecting a monitoring item, the real-time data of all sensors associated with that item is displayed below. This section presents the current monitoring data, allowing users to observe the real-time status of the sensors and their corresponding measurements. By providing this information in a clear and organized manner, the warning platform enables users to quickly assess the situation and make informed decisions. Overall, the information display window in Figure 12 offers a comprehensive view of the system's overall status, provides access to specific monitoring data through subsystems, and presents real-time data for effective monitoring and analysis.

Figure 13 shows a schematic diagram of the layout of selected underground monitoring points. The underground monitoring points mainly include ore-drawing video recognition monitoring, tunnel stress monitoring, and microseismic monitoring. The primary function of discharge video recognition monitoring is to differentiate minerals from glacial deposits based on mineral chromaticity and brightness and to estimate the approximate water content of the minerals. By identifying the types of minerals and estimating the water content, the characteristics of underground debris flows induced by glacial deposits can be preliminarily determined. The tunnel stress monitoring mainly ensures the stability of the surrounding rock in underground tunnels. The variation in stress measured by the deployed stress gauges is used to make an initial judgment on the stability of the surrounding rock. The primary purpose of microseismic monitoring is to assess the stability of rock formations based on the number and energy of microseismic events occurring within the rock layers.

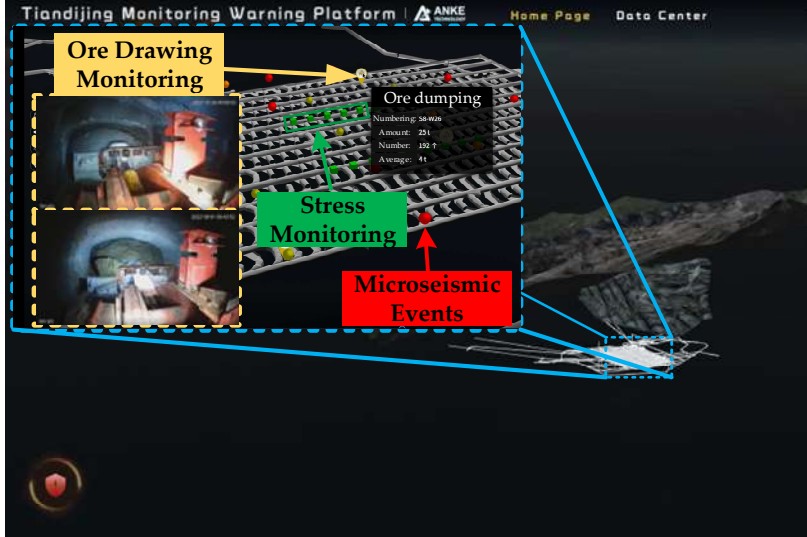

**Figure 13.** Schematic diagram of underground monitoring point layout.

4.2.2. The Basis for Early Warning Determination

Based on the aforementioned early warning monitoring system, the criteria for early warning determination include rainfall, water inflow, groundwater level, surface subsidence, pore water pressure, stress, microseismic events, and video surveillance recognition [26–30]. Here, we focus on the description of rainfall, groundwater level, surface subsidence, and microseismic monitoring.

1. Rainfall

In the study "Research on the Formation Conditions and Prevention Measures of Underground Debris Flow Induced by Natural Collapse Mining at the Pulang Copper Mine", detailed research was conducted on the rainfall that triggers debris flow disasters. The paper indicates that there is a high probability of underground debris flow occurrence when the rainfall intensity reaches 72.2 mm/h. When the rainfall intensity is 37.7 mm/h, there is a low probability of inducing underground debris flow. It is difficult for underground debris flow to form when the rainfall intensity is 23.7 mm/h. Therefore, the standard value for triggering the warning in this case is set at 37.7 mm/h. Thus, the influence of rainfall intensity on underground debris flow can be classified into three levels: Level I (less than 23.7 mm/h), Level II (between 23.7 and 37.7 mm/h), and Level III (greater than 37.7 mm/h). The corresponding rainfall intensity indices ($I_r$) for these levels are 0, 0.5, and 1, respectively.

2. Groundwater Level

The Pulang Copper Mine is located at a high altitude, and the occurrence of underground debris flow disasters is greatly influenced by glacial deposits. Based on the actual conditions, it can be inferred that the variation in the groundwater level in the mining area is related to rainfall and snowmelt. When there is a significant change in the groundwater level, the influencing factors could be either rainfall, snowmelt, or both. Therefore, there are two criteria for determining the groundwater level:

(1) When the rainfall intensity reaches the standard value (37.7 mm/h), the groundwater level index ($I_g$) is 1.

(2) When the rainfall intensity is less than 37.7 mm/h, the average groundwater level during the non-rainy season is denoted as $P_I$, the average groundwater level during the rainy season is denoted as $P_{II}$ and the historical highest groundwater level during the rainy season is denoted as $P_{III}$. The following classifications are used: if the groundwater level ($P$) is lower than $P_I$, the corresponding groundwater level index ($I_g$) is 0; if $P_I < P < P_{II}$, $I_g$ is 0.25; if $P_{II} < P < P_{III}$, $I_g$ is 0.5; if $P > P_{III}$, $I_g$ is 1.

3 Surface Subsidence

Currently, the relationship between surface subsidence and underground debris flow at the Pulang Copper Mine is still in the exploratory stage. The mechanism and actual conditions of the occurrence of underground debris flow in the rock layers, which are several hundred meters away from the surface, are difficult to determine. Therefore, the criteria for determining surface subsidence can only be based on past experience. Table 2 presents the criteria and values for displacement monitoring points based on empirical data. The warning levels are classified into four levels based on the monthly displacement measured by sensors: Level I ($d < 10$ mm), Level II ($10$ mm $\leq d < 50$ mm), Level III ($50$ mm $\leq d < 100$ mm), and Level IV ($d > 100$ mm). The corresponding surface subsidence indices ($I_d$) for these levels are 0, 0.25, 0.5, and 1, respectively.

**Table 2.** Basis for determining surface subsidence.

| Warning Level | Displacement |
|---|---|
| I | $d < 10$ mm, small displacement magnitude. |
| II | $10$ mm $\leq d < 50$ mm, uniform displacement. |
| III | $50$ mm $\leq d < 100$ mm, significant displacement. |
| IV | $d > 100$ mm, displacement discontinuity. |

The vibrational energy generated when glacial debris enters a fissure and passes through an intact ore body into a tunnel can reflect the movement characteristics of the glacial debris. Different seismic waves emitted by different seismic sources exhibit distinct wave characteristics and display different waveforms. In underground mining, microseismic blasting is commonly used, characterized by higher magnitudes and greater energy release, resulting in larger waveform amplitudes. There are noticeable differences between the waveforms of tunnel blasting and microseismic blasting. Tunnel-blasting waveforms have shorter time intervals and generate multiple relatively uniform waveforms within a short period, with each waveform being clear and relatively independent. Waveforms generated by shaft blasting generally have lower energy and propagate over shorter distances, only detectable by a few sensors near the shaft, and they do not exhibit fixed waveforms. On the other hand, microseismic waveforms are generated by rock mass failure and are characterized by relatively short durations (typically less than 400 ms). The waveform of microseismic events also varies depending on the distance from the seismic source. Therefore, in monitoring microseismic events, the focus is on distinguishing different types of microseismic events.

Based on the microseismic monitoring data collected over a period of 5 months in the shaft, collapse zone, and return airway of the Pulang Copper Mine, it was found that when the number of microseismic events ($m$) is less than 5, the occurrence of dynamic hazards is unlikely. When $5 \leq m < 10$, there is a small probability of dynamic hazards occurring. When $m \geq 10$, there is a high probability of dynamic hazards occurring. The corresponding microseismic indices ($I_m$) are 0 ($m < 5$), 0.5 ($5 \leq m < 10$), and 1 ($m \geq 10$).

### 4.2.3. Data Query

In Figure 14, the three primary options on the left-hand side window are data query, conjoint analysis, and video surveillance. Clicking on the data query button allows for querying monitoring data from sensors such as rainfall, surface displacement, hydrology, microseismicity, and stress. The conjoint analysis button generates data curves, and the video surveillance option allows for viewing real-time videos of the ore discharge point.

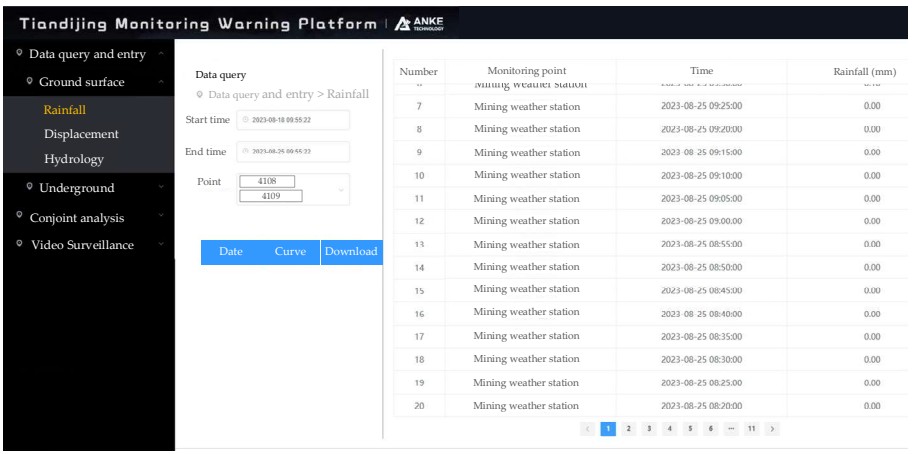

**Figure 14.** Data query system.

Taking rainfall data query as an example, in the data query window, one can select the start and end time for data retrieval. Below, the coordinates of the mining area to be queried can be entered. Clicking the data button displays real-time data in the rightmost window, consisting of data identification for the current time period, data name, time, and data value. Clicking the curve button below automatically switches to the data analysis window. Clicking the download button allows for downloading real-time data in Excel format.

*4.3. Early Warning Information Release*

The actual on-site early warning monitoring data are transmitted to the cloud. By entering the mobile phone number in the mobile phone program, the main monitoring data of the day in each monitoring area of each mine can be queried in real time. When the warning is triggered, the warning information is sent to the personal mobile device to realize the remote warning reminder.

Taking stress warning and microseismic warning as examples, Figure 15 presents the warning information sent to the responsible team members in the underground debris flow monitoring and warning system. The information includes the monitoring area, warning level, monitoring point information, stress data, time of microseismic events, and their energy. Since the deployment of this warning system, with the active cooperation between the warning monitoring system and the mining team, no underground debris flow events have occurred in the monitored area.

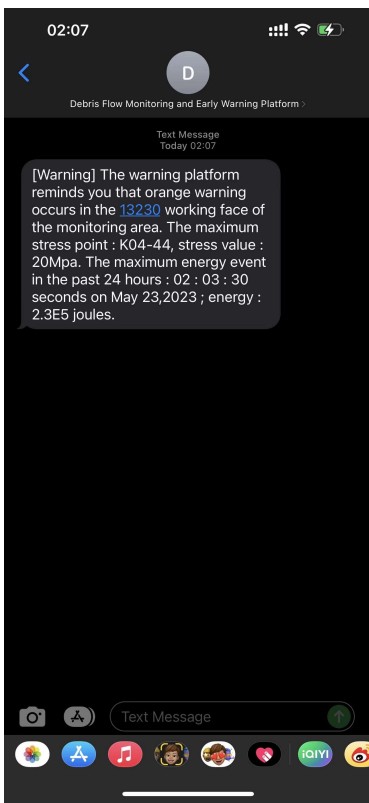

**Figure 15.** Release of warning information.

## 5. Conclusions

The traditional underground debris flow monitoring methods in the past were primarily focused on hydrological monitoring, such as monitoring rainfall and groundwater levels. However, the approach introduced in this paper, known as the "weather-surface-underground" approach, presents a more comprehensive monitoring method. This approach overcomes the limitations of traditional methods by integrating and analyzing various monitoring data to make comprehensive judgments and trigger alerts when necessary.

To improve the accuracy and reliability of the monitoring system, the paper proposes a data preprocessing method for key monitoring data, including rainfall, inflow, groundwater levels, and displacement. This preprocessing method helps reduce errors caused by data inaccuracies and minimizes the occurrence of false alarms. Furthermore, the paper introduces a comprehensive alert determination algorithm that integrates weather monitoring indicators, surface monitoring indicators, and underground detection indicators. By considering multiple indicators instead of relying on a single indicator, this algorithm helps avoid errors that may arise from single indicator alerts. The paper also presents an alarm information management system that accurately and efficiently sends alert information to relevant personnel. This system distinguishes between comprehensive alerts, which require immediate attention, and single alerts, which may not be as critical. By differentiating between these types of alerts, the system reduces the overall cost of alerts and improves alert efficiency.

The effectiveness of the proposed monitoring system and algorithms has been demonstrated through their successful application at the Pulang Copper Mine, where they have yielded positive monitoring results. Overall, the paper introduces a comprehensive approach to underground debris flow monitoring that integrates various monitoring data, employs preprocessing techniques to enhance data accuracy, utilizes a comprehensive alert determination algorithm, and incorporates an efficient alarm information management system. These advancements contribute to improved monitoring accuracy, reduced false alarms, and enhanced alert efficiency in the mining industry.

**Author Contributions:** Conceptualization, Q.Z. and S.Z.; methodology, A.W.; validation, Z.L., Y.S., S.W., M.F. and X.Q.; formal analysis, A.W.; investigation, Q.Z.; resources, S.Z.; data curation, Y.S.; writing—original draft preparation, M.W.; writing—review and editing, M.W.; visualization, S.W.; supervision, A.W.; project administration, S.Z.; funding acquisition, S.Z. All authors have read and agreed to the published version of the manuscript.

**Funding:** This work was supported by the National Key Research and Development Program of China (No. 2022YFC2903803); the Young Talents Lifting Project of China Association for Science and Technology (No. 2021QNRC001); the Major Science and Technology Innovation Project of Shandong Province (No.2019SDZY02).

**Institutional Review Board Statement:** Not applicable.

**Informed Consent Statement:** Not applicable.

**Data Availability Statement:** The data used to support the findings of this study are available from the corresponding author upon request.

**Conflicts of Interest:** The authors declare no conflict of interest.

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
