# Peer review of "Research on Real-Time Monitoring and Warning Technology for Multi-Parameter Underground Debris Flow"

_sustainability, doi:10.3390/su152015006_

Round 1
Reviewer 1 Report
Please immediately revise or correct your manuscript according to the suggestions for improvement that I wrote in your manuscript!

Please immediately revise or correct your manuscript according to the suggestions for improvement that I wrote in your manuscript!
Author Response
Dear Professor,
I sincerely appreciate your valuable feedback on my article. I have revised all 30 suggestions you made in the original article, including the summary and introduction. I have also included detailed explanations of formulas and images in the text, further enhancing the novelty of the article and adding references.
Reviewer 2 Report
Reviewer Comments for Research on Real-Time Monitoring and Warning Technology for Multi Parameter Underground Debris Flow
This manuscript presents a practical and feasible underground debris flow monitoring and warning system is designed by onsite monitoring, establishing monitoring models, setting judgment criteria, and integrating multiple preprocessing parameters. In this manuscript, the key data such as rainfall monitoring, groundwater monitoring and displacement monitoring are preprocessed to improve the effectiveness of monitoring data and reduce the frequency of data fluctuation. The manuscript is well organized and well-rounded, but it needs major modifications before acceptance. The following comments may be helpful for the authors to refine this manuscript:
1. In Section 1.2, most of the second paragraph talks about the theoretical basis of deep well precipitation. This part should belong to the ‘Introduction’.
2. In Section 1.3, ‘A large number of practices show that the data after soft threshold processing is smoother and more regular. In this paper, soft threshold processing is selected for noise reduction research.’ References are needed to support this conclusion.
3. In Section 2.2, ‘The weight can be determined according to the analysis of the influencing factors of debris flow disasters in the region and the activity of monitoring indicators, so as to improve the adaptability of the early warning algorithm to the complex environment.’ The authors need to give more details about how to determine these weights.
4. In Section 3.1.1, ‘The mining area monitoring model is established through indoor modeling and onsite field reconnaissance, and imported into the big data platform (Figure 8). The three-dimensional model is divided into sky model, ground model and underground model.’ The description and figure of the model are not clear enough to understand the whole model. More details must be added.
5. In Section 3.3, ‘The data query system is shown in Figure 12, which includes five subsystems: stress, microseismic, hydrology, surface subsidence and ore drawing. You can select a certain measuring point in a certain area within a certain time period to query, or you can choose to view specific data or generate data curves.’ Instead of just showing a simple introduction to the system, the authors need to provide technical methods and results.
6. In Section 3.4 Early warning information release, how is the early warning information generated inside the system? What kind of analysis and decision-making methods are used?
Minor editing of English language required
Author Response
Dear Professor:
Thank you very much for your valuable suggestions. I have learned a lot from them and addressed my shortcomings. Regarding the six professional opinions you provided, I have made detailed modifications in the document. Here is my response to each of them:
1、In Section 1.2, most of the second paragraph talks about the theoretical basis of deep well precipitation. This part should belong to the ‘Introduction’.
Answer1:The reason why the theory of deep-seated precipitation was not included in the introduction section is that the theory serves as a means to predict groundwater levels, and I believe it should be elaborated on in detail here.
2、In Section 1.3, ‘A large number of practices show that the data after soft threshold processing is smoother and more regular. In this paper, soft threshold processing is selected for noise reduction research.’ References are needed to support this conclusion.
Answer2:I have made additions to the text based on your suggestions.
3、In Section 2.2, ‘The weight can be determined according to the analysis of the influencing factors of debris flow disasters in the region and the activity of monitoring indicators, so as to improve the adaptability of the early warning algorithm to the complex environment.’ The authors need to give more details about how to determine these weights.
Answer3:I have elaborated on the warning index in the practical application section of Chapter 3 based on your suggestions.
4、In Section 3.1.1, ‘The mining area monitoring model is established through indoor modeling and onsite field reconnaissance, and imported into the big data platform (Figure 8). The three-dimensional model is divided into sky model, ground model and underground model.’ The description and figure of the model are not clear enough to understand the whole model. More details must be added.
Answer4:The 3D model includes the sky model, the surface model, and the underground model. When selecting images, the models were intentionally scaled down to display comprehensively. Regarding the textual description of the models, I have provided detailed supplementary explanations based on your suggestions.
5、In Section 3.3, ‘The data query system is shown in Figure 12, which includes five subsystems: stress, microseismic, hydrology, surface subsidence and ore drawing. You can select a certain measuring point in a certain area within a certain time period to query, or you can choose to view specific data or generate data curves.’ Instead of just showing a simple introduction to the system, the authors need to provide technical methods and results.
Answer5:Whenever a single warning is triggered, relevant personnel will go to the warning area to handle it. Therefore, since the application of the warning system, there have been no underground debris flow incidents in the mining area, and there is a lack of data triggering comprehensive warnings.
6、In Section 3.4 Early warning information release, how is the early warning information generated inside the system? What kind of analysis and decision-making methods are used?
Answer6:I have made additions to the text based on your suggestions.
Reviewer 3 Report
1)On page 2, lines 67-70, there is an error in the description. Please correct or provide references.
2)The core of this study is the fusion and discrimination methods of rainfall monitoring, groundwater monitoring, displacement monitoring, and other data, as well as the threshold research for triggering early warning. However, the description of key issues is too few, which is difficult to understand.
3)The overall description of the manuscript is not concise and has high redundancy.
4)The language description needs further refinement and correction.
Extensive editing of English language required
Author Response
Dear Professor:
Thank you very much for your valuable 4 suggestions. Through your suggestions, I have gained a deep understanding of the shortcomings of my article, and I have made comprehensive and detailed revisions to the article.
Thank you once again for your valuable suggestions!
Round 2
Reviewer 1 Report
Please shorten the abstract writing, pay attention to clarity between sections of the article, especially between methods, results and discussion!
Please make more references, not just the 30 articles used as references!

Please shorten the abstract writing, pay attention to clarity between sections of the article, especially between methods, results and discussion!
Please make more references, not just the 30 articles used as references!
Author Response
Dear Reviewer,
First of all, thank you very much for your valuable feedback on my article. I have made the following modifications based on your suggestions:
- I have condensed the abstract section to highlight the methodology, results, and discussions.
- I have refined the language in certain parts of the article.
- I have thoroughly checked the relevance of the references and removed some of them.
Thank you once again for your valuable input!
Best regards
Reviewer 2 Report
All recommendations have been appropriately modified.
Author Response
Dear Reviewer,
Thank you very much for your valuable feedback on my article. This time, I have made the following modifications to the article:
- I condensed the summary section to highlight the methodology, results, and discussion.
- I improved the language in some parts of the article.
- I have thoroughly checked the relevance of the references and removed some of them.
Thank you again for your valuable comments!
With respect
Reviewer 3 Report
The overall quality of the revised version has greatly improved. Please organize the final version format of the manuscript according to the requirements of the journal, and it is recommended to receive and publish it.
Minor editing of English language required.
Author Response

(The authors gave the same response as above.)
